# Genetic Diversity and Connectivity of the Vulnerable Species *Phengaris nausithous* in Palencia (Northern Spain) [note 1]

**DOI:** 10.3390/insects16020193

**Published:** 2025-02-11

**Authors:** Luis Fernando Sánchez-Sastre, Óscar Ramírez-del-Palacio, Pablo Martín-Ramos, María-Ángeles Hernández-Minguillón

**Affiliations:** 1Área de Ingeniería Cartográfica, Geodésica y Fotogrametría, ETSIIAA, Universidad de Valladolid, Avenida de Madrid 44, 34004 Palencia, Spain; luisfernando.sanchez@uva.es; 2Departamento de Ingeniería Agrícola y Forestal, ETSIIAA, Universidad de Valladolid, Avenida de Madrid 44, 34004 Palencia, Spain; o.ramirezdelpalacio@gmail.com; 3Departamento de Biología Ambiental, Facultad de Ciencias, Universidad de Navarra, 31080 Pamplona, Spain; mahermin@unav.es

**Keywords:** habitat fragmentation, population genetics, conservation biology, ecological connectivity, endangered species, gene flow, metapopulation dynamics

## Abstract

Butterflies worldwide are experiencing significant population declines, largely due to habitat loss and fragmentation. This is particularly concerning for species with specific habitat requirements, like the dusky large blue butterfly, which depends on a single plant species and specific ant species to complete its life cycle. In northern Spain, we studied several populations of this vulnerable butterfly to understand how well they can move and maintain genetic connections between fragmented habitats. By analyzing DNA samples from butterfly wings, we discovered that most populations were genetically isolated from each other, except for two populations within a natural park that showed strong connectivity. This suggests that the presence of suitable habitats between populations, acting as “stepping stones”, can help maintain genetic exchange. Our findings provide valuable insights for conservation efforts, highlighting the importance of preserving and connecting suitable habitats to ensure the long-term survival of this threatened species. This study also revealed that what was previously thought to be separate populations within the natural park actually function as a single, interconnected population.

## 1. Introduction

Habitat loss and/or fragmentation is one of the main threats to biodiversity conservation [1,2], resulting in an increasing number of species being forced to survive in fragmented landscapes [3,4]. Habitat fragmentation occurs when a large continuous area is divided into smaller, isolated patches by a matrix of habitats different from the original habitat [5], leading to reduced connectivity [6].

Within the context of a general insect decline reported by various studies [7,8], butterflies have also experienced a decline observed in recent decades [9,10]. Butterflies are considered sensitive to environmental changes [11], specifically to alterations in local conditions, food sources, or host plant availability [12,13], with habitat loss due to natural events or human activities being the primary cause [14,15]. In this regard, over recent decades, several butterfly species have been used as models for studying the effects of habitat loss, fragmentation, and degradation at different spatial scales [16,17]. Moreover, it is significant that approximately 40% of grassland butterflies are distributed in metapopulations [18], that is, in a set of local populations that interact through migration and gene flow, whose persistence over time and dynamics are influenced by variations in habitat and connectivity [17,19]. Metapopulation dynamics focus on local extinction and recolonization processes, just as population dynamics deal with the births and deaths of the individuals that compose it [4,20].

Small metapopulations occupying patches with low connectivity have a low probability of continuous occupation [21]. In these populations, it is logical to expect few emigrants, but their presence can still facilitate connectivity between populations and occasional movement of individuals between more distant populations occasional movement of individuals between more distant populations [22].

Another relevant aspect associated with small metapopulations and high local isolation is that they are subject to a high risk of stochastic extinction [23], and this extinction is exacerbated by the limited opportunity to recolonize the fragmented habitats in which they are distributed [24]. Furthermore, the high specificity of some species living in isolated populations makes them more vulnerable [25].

Therefore, connectivity promotes genetic diversity among small local populations that would otherwise be affected by genetic drift [26]. A potential consequence of reduced genetic diversity would be low adaptability to changes in environmental conditions [27]. Moreover, genetic factors are important when populations are in decline, as reductions in levels of genetic variation suggest factors that frequently play a significant role in population extinction [28].

The dusky large blue *Phengaris nausithous* (Bergsteässer, 1779) is a small butterfly of the family Lycaenidae with a wingspan of 30–36 mm [29] that depends on the presence of its host plant *Sanguisorba officinalis* L. and host ants *Myrmica scabrinodis* (Nylander, 1846) [30] and *Myrmica rubra* (Linnaeus, 1758) [31,32] for larval development through a cuckoo strategy. The host plant is perennial and hydrophilic, and is sensitive to changes in grassland management and fragmentation [33].

*Phengaris nausithous* is distributed across Spain, France, and Central Europe to Asia [34]. In Spain, it is primarily limited to areas of the mountainous belt of the Castilla y León region (Palencia, León, Burgos, and Soria), although it has also been reported in the provinces of Asturias, Cantabria, Guadalajara, and Madrid [35,36,37,38] (Figure 1).

Globally, it is cataloged by the IUCN Red List as near threatened and is strictly protected and listed in Annexes II and IV of the Habitats Directive (Council Directive 92/43/EEC) and Annex II of the Bern Convention [39]. In Spain, it is categorized as ‘vulnerable’ in the Spanish Catalog of Threatened Species [40]. Its habitat, consisting of lowland hay meadows, is of community interest (code 6510 of the Habitats Directive). The main threats to the species are habitat loss and fragmentation [41] and the lack of host plant availability during the flight period [37]. Additionally, climate change may influence modifications in the present and future distribution range of the species [42].

To date, studies on *P. nausithous* in Europe have focused on analyzing its distribution, ecology, and phylogeny [43,44,45,46]. Although genetic studies on this species are not numerous, we can highlight the work of Figurny-Puchalska et al. [23], who analyzed the population genetic structure of *P. nausithous* and *Phengaris teleius* (Virgstrassir, 1779) in Poland and Russia, as well as the study by Hollós et al. [47], who compared morphometric differentiation and genetic patterns of *P. nausithous* in Hungary and Romania. Zeisset et al. [48] designed the original nuclear microsatellites for *P. nausithous*, which have enabled the study of the population structure of *P. nausithous* and the parasitoid *Neotypus melanocephalus* (Gmelin, 1790) in a fragmented landscape [49], and Ritter et al. [50] conducted a phylogenetic study for *P. nausithous* and *P. teleius*. These microsatellites were also used in studies of other *Phengaris* species [51,52,53,54,55,56,57].

Until the beginning of this work in 2019, only eight habitat patches occupied by *P. nausithous* were known in the westernmost part of Montaña Palentina (Carrión River watershed) in contrast to the 26 populations in the eastern part (for simplicity, the term population will occasionally be used to designate habitat patch(es) with the presence of *P. nausithous*). Based on a maximum recorded displacement of 5 km observed for the species [58], Jubete and Román [41] grouped all these populations into five metapopulations, considering all subpopulations at a distance of less than 5 km as part of the same metapopulation.

The objective of the present study was, firstly, to improve knowledge about the distribution of *P. nausithous* and, secondly, to understand the genetic structure and degree of connectivity among several geographically proximate populations (2–43 km) of this species in the Montaña Palentina area (northern Spain), as well as the potential barriers presented by the fragmentation matrix of its habitat and its possible genetic isolation from an external population (Soria) located approximately 200 km away.

## 2. Materials and Methods

### 2.1. Study Area

The fieldwork was primarily conducted in the Cantabrian Mountains (northern Spain), specifically within the Montaña Palentina Natural Park (Easting 375608 m, Northing 4755047 m, EPSG 25830) and several nearby areas (Figure 1). Within this protected area, two major rivers, the Carrión and the Pisuerga, originate. Within the park, work was mainly conducted in its westernmost and most rugged part, the Carrión River watershed, which, due to its orography, was the least studied area to date in contrast to the eastern part, corresponding to the Pisuerga River watershed, which is characterized by gentler orography and wider valleys (Figure 2). Two population patches from Soria province (at a distance of about 200 km from the main study area in Palencia; see Figure 1) were included as external populations. Two areas from the eastern Leonese mountains adjacent to the Palentine mountains and a dusky large blue butterfly reserve near the southeastern boundary of the natural park were also included.

The Montaña Palentina is a mountainous area with small valley bottoms dedicated to pastures, most of which are used for both mowing and grazing, with a mean annual temperature of 8.3 °C and 1180 mm of accumulated annual precipitation. Mowing generally begins in late June, directly interfering with the emergence of *P. nausithous*, which begins its flight period in early July. In the studied area, habitat patch sizes range between 0.2 and 18 ha (most below 3 ha) in the Carrión area and between 0.6 and 28 ha in the Pisuerga River watershed. Pasture areas in the lower parts of the valleys are bordered by extensive shrubland and conifer masses, with oak and beech forests appearing in the eastern part. The habitat matrix for *P. nausithous* is completed by the presence of reservoirs on the Carrión River in the western part and on the Pisuerga River in the eastern part (Figure 2).

Regarding population sizes in the area, in 2020, a capture–mark–recapture (CMR) campaign was conducted on two nearby populations (2.5 km) with notable host plant density but different management characteristics and sizes. For the first population (10.5 ha used for mowing and grazing), an estimate of 202 individuals with a mean lifespan of 2.3 days was obtained, and for the second (0.5 ha unexploited), 107 individuals with a mean lifespan of 3.3 days were estimated, with some individuals in both populations surviving up to 12 days as determined through recapture [59,60].

### 2.2. Study Area Survey

Over 60 field visits were conducted between June and September of 2019 and 2020 with the aim of better understanding the presence of *P. nausithous* in the westernmost and most rugged area of Montaña Palentina, as mentioned above, in contrast to the more studied eastern area corresponding to the Pisuerga River watershed. This aimed to expand knowledge about the species’ actual distribution within the entire study area to subsequently select sampling points for genetic analysis. During each campaign, we covered approximately 500 km and directly inspected an approximate surface area of 500 ha of terrain. In addition to *P. nausithous* populations, all detected presence of its host plant was mapped whether the butterfly was present or not. Sites with suitable habitat characteristics and the presence of the host plant but without butterfly occurrence were labeled as POT1-POT3 (Figure 2), representing potential habitat patches for future colonization.

### 2.3. Genetic Study

#### 2.3.1. Sampling Point Selection

In 2020 and 2021, 10 habitat patches with *P. nausithous* presence were sampled and grouped into 6 zones or populations (5 in Montaña Palentina and surrounding area, and 1 in Soria province) based on their geographical location, population size, and potential impacts to optimize sample collection with available resources (Table 1). Thus, the following zones/populations were established:PSW: Population southwest of the Natural Park. One patch was sampled;PNW: Population in the Carrión River watershed and part of the eastern Leonese mountains. Located west of Camporredondo and Compuerto reservoirs: three patches were sampled 2.3–10 km apart;PC: Population in the Carrión River watershed but east of Camporredondo and Compuerto reservoirs: two patches were sampled 8 km apart;PNE: Population in the Pisuerga River watershed: one patch was sampled 11 km east of the nearest PC patch;PSE: Reserve southeast of the Natural Park: one patch was sampled at a distance of 17.8 km south of PNE;Soria: Samples were taken from two patches 2.6 km apart in Soria province, approximately 200 km from the previous five.

The remaining known points with *P. nausithous* presence have been grouped into populations or zones P1–P6 (Table 2):P1: Groups two patches occupied by the butterfly and close to PNW;P2: Groups 4 nearby patches located between P1 and PC;P3: Patch extending along a valley bottom between P1 and PC, south of P2;P4: Small patch with few individuals on the northeastern border of the Carrión Valley;P5: Large patch located right at the watershed divide between Carrión and Pisuerga;P6: Zone grouping all patches occupied by *P. nausithous* in the Pisuerga River watershed, which would also include PNE.

This results in a situation as shown in Figure 2, which displays all populations or zones known to date with *P. nausithous* presence in Montaña Palentina, as well as those used for genetic sampling. Additionally, potential habitat areas without butterfly presence are shown.

#### 2.3.2. DNA Sampling and Extraction

For each sampling at each of the 10 sites, 1 to 4 imagos were captured using an entomological net. Once captured, each specimen was handled using precision entomological forceps, and a non-lethal wing-clip (2–3 mm^2^ of tissue) was performed to obtain tissue samples that were stored in 95% ethanol, releasing the individual afterward. The impact of this technique is similar to damage produced by bird attacks or general abrasion that can be observed in many lepidopterans in the field after emergence. It is important to note that flight capacity is not reduced and that, although there may be certain short-term behavioral changes associated with capture, survival rates are not affected [61,62,63].

Sampling events were spaced at least 3 days apart at each site to respect the average lifespan of 2.7 days of imagos [64,65], thus trying to minimize population impact and avoid recapturing specimens.

Sampling was conducted during the flight period from 8 July to 10 August 2020 and from 15 July to 6 August 2021.

DNA was extracted from wing fragments obtained in the field using the Genomic kit (NucleoSpin Tissue; Macherey-Nagel, Dueren, Germany) and following the tissue protocol as described by Rutkowski et al. [51].

Eight microsatellite loci were tested, six previously designed by [48] for *P. nausithous* (Macu5, Macu8, Macu9, Macu11, Macu16, and Macu17) and two by [53] for *Phengaris arion* (Linnaeus, 1758) (Macu26 and Macu44). Five (Macu5, Macu8, Macu9, Macu11, and Macu44) were amplified with an annealing temperature of 60 °C, Macu16 and Macu17 were amplified at 63 °C, and Macu26 was amplified at 57 °C. Of these loci, five (Macu5, Macu8, Macu11, Macu16, and Macu44) were successfully amplified for a sufficient number of specimens from the analyzed populations. The remaining ones (Macu9, Macu17, and Macu26) were discarded.

PCR conditions were standardized (Primers 5 µM, Buffer 10×, MgCl_2_ 25 mM, dNTPs 2 mM, TaqGreen, and DNA 4 µL with a final reaction volume of 25 µL). Amplification was performed in a T100 Thermal Cycler (Biorad, Hercules, CA, USA), with an initial activation at 95 °C for 4 min, followed by 30 cycles × 95 °C for 30 min. The annealing temperature for each primer was as indicated above for 30 s, followed by 72 °C for 60 s and a final extension of 10 min at 72 °C. After purification with the ThermoScientific GeneJET Kit (Thermo Fisher Scientific, Waltham, MA, USA), they were genotyped using a 3730XL DNA analyzer from Applied Biosystems (Waltham, MA, USA) and analyzed with Peak Scanner Software v.1.0, also from Applied Biosystems.

#### 2.3.3. Statistical Analyses

The GENETIX 4.05 program [66] was used to calculate the mean number of alleles per population, observed and expected heterozygosity [67], genetic distance, inbreeding coefficient (*F*is) [68], and gene flow between populations (*N*m) [69]. An *F*is value of 0 indicates no inbreeding, *F*is = 1 indicates complete inbreeding, and *F*is = −1 occurs when all individuals are heterozygous. The *N*m value ranges between 0 and 1 when gene flow is low; thus, *N*m > 1 indicates high flow, and *N*m > 4 indicates that the population is panmictic [70].

Analysis to determine deviation from Hardy–Weinberg equilibrium (HWE) and locating patterns of linkage disequilibrium in a population was performed using ARLEQUIN 3.5 [71], followed by Bonferroni correction for multiple comparisons [72].

The occurrence of null alleles, allele dropout, and stutter bands was assessed using MICRO-CHECKER 2.2.0.3 [73]. Allelic richness (*Ar*), which is the index of the number of alleles corrected for sample size using rarefaction, and private allelic richness, which is the measure that shows how much a population differs from other populations, were calculated using the HP-RARE program [74] for the first and second level. The smallest size to obtain *Ar* was 22 alleles.

The Bayesian admixture analyses were performed with STRUCTURE 2.3.4 [75]. Two tests were conducted. In the first analysis, data from the six populations were tested for *K* = 1–6. In the second analysis, given the result, the Soria population was excluded (*K* = 1–5). In both cases, 10 different runs were executed for each *K* to estimate mean values and standard deviation with a burn-in period of 500,000 and 1,000,000 Markov Chain Monte Carlo replications under the assumption of admixture and correlated allele frequencies. The program was first run without using information about the butterflies’ places of origin to find the most probable number of groups, which was also assessed using the ad hoc statistic Δ*K* proposed by Evanno et al. [76]. This procedure was implemented using the StructureSelector website [77].

The relationship between genetic distances and geographical distances (corresponding to the Euclidean distance between two sampled populations) allows for an understanding of possible isolation by distance (IBD), which was tested with the non-parametric Mantel test using the FSTAT program [78].

The BOTTLENECK 1.2 program [79] was used to find heterozygosity excess in a population at mutation-drift equilibrium, which shows if there has been a recent bottleneck in the population. The Infinite Allele Model (IAM) and Stepwise Mutation Model (SMM) in a bottlenecked population were used to perform the Wilcoxon one-tailed test, considered a more powerful approach when having fewer than 20 loci [80]. In all cases, 10,000 simulation replicates were performed.

The effective population size (*Ne*) was estimated using the linkage disequilibrium method employed in LDNe 1.31 software [81]. If the estimated values of *Ne* are negative or infinite, they are interpreted as an infinite estimate, meaning there is no evidence of any disequilibrium caused by genetic drift due to a possible finite number of parents, so everything can be explained by the need for a larger sample size.

First-generation migrants were identified using GENECLASS2 [82]. The likelihood-based test statistic was calculated by comparing the marginal probability of individual multilocus genotypes with the distribution of randomly generated multilocus genotypes (10,000 replicates) using the method of Paetkau et al. [83]. An alpha level of 0.01 was used to determine critical values, with a frequency of 0.1 assigned for missing alleles.

The spatial pattern of genetic variation was analyzed using Moran’s I statistic [84] in GeoDa v.1.22.0.14 [85] with randomization and 999 permutations to test whether the spatial distribution of genetic variation differed from random.

## 3. Results

### 3.1. Study Area Survey

During 2019 and 2020, 15 new habitat patches occupied by *P. nausithous* were detected, and the area of 3 previously known patches was expanded, totaling 21 sites with dusky large blue presence in the Carrión River watershed [60,86]. Similarly, a new site with dusky large blue presence was located at the watershed divide between the two rivers (P5; see Figure 2). Additionally, during fieldwork, several areas with the presence of the host plant but without dusky large blue were identified.

Regarding UTM grid squares, the 100 km^2^ square 30TUN54 was added, along with 27 new 1 km^2^ UTM cells (access to these data is subject to prior consultation with the Montaña Palentina Natural Park management authority).

Flight periods within the study area were 52 and 50 days, respectively, from 10 July to 29 August 2019 and from 6 July to 26 August 2020.

### 3.2. Genetic Study

A total of 111 *P. nausithous* individuals were analyzed, 86 in 2020 and 25 in 2021. In the six populations, 42 alleles were found at five loci with a mean value of 3.30. In no population was heterozygosity above the expected level. No linkage disequilibrium was found between pairs of loci for each population after performing the Bonferroni correction for multiple tests. The mean frequency of null alleles per locus across all populations was less than 4% (99% CI). Regarding the inbreeding value (*F*is), it was above 0 in four populations and negative in PSE. The mean number of alleles (MNA) and allelic richness (*Ar*) showed the highest values in the Soria population (Table 3). Private allelic richness ranged from 0.01 in PC to 2.97 in Soria.

Genetic distance was high between the Soria population and the remaining ones except for PSE. Gene flow was low (*N*m ≤ 1) between Soria and four populations, with the highest flow between PNW and PC populations (*N*m = 4.60); as *N*m > 4, both populations can be considered panmictic (Table 4).

The ad hoc statistic Δ*K* showed strongest support for *K* = 2, followed by a secondary peak at *K* = 5 (Figure 3A). Visualizing the STRUCTURE plots, there was a clear difference between specimens from the Soria populations and the other five populations. Secondarily, four populations were observed: PSW, PNE, and PSE, plus the combination of PNW and PC (Figure 4A).

From the analysis performed among the five populations, after excluding Soria data, a genetic structure model of *K* = 4 was obtained (Figure 3B). In this structure, PNW and PC appeared differentiated as a single unit (hereafter referred to as PNWPC), along with the other three populations: PSW, PNE, and PSE (Figure 4B).

The estimated population size for each population was 18 (CI = 5–infinite) for Soria, 33 (CI = 10–infinite) for PSW, 9 (CI = 1–infinite) for PNW, 234 (CI = 2–infinite) for PC, −20 (CI = 2–infinite) for PNE, and 8 (CI = 1–infinite) for PSE. The bottleneck analysis showed no evidence of a reduction in the populations.

In the IBD (isolation by distance) analysis for butterflies from all six populations, the Mantel test (FSAT program) showed evidence of IBD (R^2^ = 0.61, *p* = 0.017) with evidence of two groups, one corresponding to Soria specimens, and the other formed by the other five populations. When the Mantel test was repeated, excluding Soria specimens, the IBD was weak and non-significant (R^2^ = 0.034, *p* = 0.698).

The first-generation migrant analysis identified one migrant (1.38% migration rate per generation) from the PC population captured in the PNW population, supporting the genetic connectivity between these populations. Moran’s I analysis rejected the null hypothesis of spatial randomness (*p* = 0.001, z = 6.1982), indicating that the spatial distribution of genetic variation follows a non-random pattern.

## 4. Discussion

*Phengaris nausithous* has a discontinuous distribution across Europe. While some studies suggested that in their study areas, it could be considered cryptic species with divergence in the mitochondrial COI gene [43] or different species or subspecies [47], Ritter et al. [50], analyzing samples from within the Eurasian distribution area, concluded that these are not necessarily cryptic species but rather reflect *Wolbachia* infection and phylogeographic structure. These authors included three specimens from Spain, indicating that the Iberian Peninsula would be one of the glacial refugia for *P. nausithous* in Europe, along with the Italian and Balkan peninsulas. From this, we can assume that the specimens analyzed here correspond to the species that extends across Europe and Asia.

Hollós et al. [47] genetically analyzed two populations 800 km apart, one in Transylvania (Romania) and another in Örség (Hungary), and observed that both populations were genetically separated into two clusters. Our results also indicate a genetic structure of *K* = 2 when considering the six analyzed populations. The Soria population, at a distance of about 200 km from northern Palencia, would be genetically isolated by distance, as shown by the Mantel test results.

In northern Palencia, four populations are genetically differentiated. Among the populations in the upper Carrión River area, PNW and PC show no genetic structure, suggesting a metapopulation formed by all patches occupied by the dusky large blue butterfly in PNW and PC. Zones P1, P2, and P3, where the species presence has been confirmed, although pending genetic analysis confirmation, are likely part of the same metapopulation due to their location, serving as links between the genetically analyzed PNW and PC zones.

Furthermore, three other clusters were differentiated, corresponding to PSW, PNE, and PSE. The populations defined as PSW and PNE are the most isolated of all those considered in this study. PNE is the only population patch corresponding to the eastern part of Montaña Palentina that could be included in P6.

Two key factors from the literature may explain these results: species biology and habitat matrix typology. In this regard, the importance of landscape and habitat matrix in the occurrence, density, and dispersal of species such as *P. nausithous* has been noted by many studies [46,87,88].

Generally, *P. nausithous* shows habitat fidelity and low individual exchange between patches due to its limited movement capacity, between 80 and 400 m [58,64,65], with the maximum recorded distance being 5.1 km [58]. Therefore, the distance between population patches relative to flight distance and survival rate would hinder movement between them [64].

In our study, the populations defined as PSW and PNE were the most isolated of all those considered, making this result expected. PSW is about 13 km away from the nearest part of PNW with no known intermediate population patches (Table 5). PSE is 9 km south of P6’s distribution, and it also has no known intermediate colonies. PNW and PSW are nearly 40 km apart and separated by a crop matrix where only a few small patches with scarce individuals have been found [38]. PNE is the only population patch corresponding to the eastern zone of Montaña Palentina that could be included in P6 and remains relatively distant from the western part, with barely any habitat patches between them (P4 and P5), which may hinder the connection between populations of the two watersheds.

Furthermore, Skórka et al. [22] noted that *P. teleius*, although reluctant to cross patch edges when doing so, could cover relatively long and direct distances, spending less time resting, foraging, and ovipositing. Nowicki et al. [87] indicated that for *P. teleius* and *P. nausithous*, although an inhospitable matrix may create strong selection against dispersal, resulting in very low emigration rates, there could be selection for individuals with good dispersal capabilities that undertake long-distance flights. Additionally, Pérez-Sánchez et al. [46] found that a high percentage of forest cover in the matrix positively affects *P. nausithous* presence, and habitat patches within the forest near small streams can serve as stepping stones. Similarly, Villemey et al. [89] concluded that for specialist species with low dispersal capacity, a mosaic of grassland patches and woody habitats might be more effective than improving linear connectivity. Also, authors such as Batáry et al. [90] indicate that *P. nausithous* shows a greater preference for forest edges, and Ugelvig et al. [53] suggest that for *P. arion*, the presence of *S. officinalis* in areas near river banks and reservoirs could facilitate connectivity between populations, also acting as stepping stones.

In our study area, both the zone of PNW and PC populations and the unstudied eastern zone present a configuration of valley bottoms with pastures surrounded by forest and scrubland and crossed by small streams and rivers, with habitat patches not far from each other. In many of the habitat patches subject to livestock activity, *S. officinalis* persists on edges, small wetland areas, or stream banks. This landscape configuration, despite physical barriers such as reservoirs and anthropogenic disturbances from early mowing, seems to allow metapopulation functioning in the westernmost part of Montaña Palentina, as occurs in other parts of Europe [49].

Regarding the genetic variability of *P. nausithous*, it is low in the studied Cantabrian area (*Ar* = 2.4 to 3.8) compared to the Soria population (*Ar* = 5.4), despite the smaller number of specimens and sampled area in Soria. This variability is also lower than that found in Germany (*Ar* = 3.5 to 5.5) [49]. Population size and isolation are key factors influencing genetic variability. In our study, population size does not appear to be one of the factors, as the highest genetic variability corresponds to Soria (*Ar* = 5.4) with an estimated number of *Ne* = 18, while the PC population with *Ne* = 234 shows the lowest genetic variability (*Ar* = 2.24). Isolation could be an influential factor, as it affects species with poor dispersal abilities. This could be exacerbated by early mowing since, although this activity may promote individual dispersal, it can also have a negative effect if most individuals in the patch are not well adapted to moving through the matrix and there are no other patches sufficiently nearby [91]. In any case, low genetic variability increases extinction risk in butterflies due to decreased adaptability [92]. Thus, Sielezniew and Rutkowski [52] indicated the need for the protection of *P. arion* in Poland to prevent a reduction in genetic variability.

Nevertheless, despite the small area occupied by the studied populations and the few and small patches found between them, from a genetic perspective and considering their structure and connectivity, no bottleneck indicating extinction danger has been found in the study area, suggesting that sporadic exchanges between populations may occur.

Such small and isolated populations are often considered “living dead”—bound for extinction due to their high risk of stochastic extinctions [93]. Although such populations are generally considered to be of low conservation value, recent research on other *Phengaris* species [56] has shown that their existence can still enhance species survival at the landscape scale by improving connectivity between other populations and facilitating gene flow, even if limited.

Considering these findings in a broader conservation context, the situation of *P. nausithous* in Montaña Palentina presents both challenges and opportunities. While habitat fragmentation and low genetic variability pose risks, the existence of functional metapopulations demonstrates the species’ capacity to maintain genetic connectivity through appropriate habitat matrices. The contrast between connected populations in areas with suitable stepping-stone habitats and isolated populations in more fragmented landscapes emphasizes the importance of landscape-scale conservation planning. Traditional management practices like mowing, when appropriately timed, could play a crucial role in maintaining habitat quality while allowing for successful reproduction. Furthermore, the identification of stepping-stone habitats provides clear targets for conservation efforts aimed at maintaining and enhancing population connectivity. These insights can inform evidence-based conservation strategies not only for *P. nausithous* but also for other specialized butterfly species in fragmented landscapes.

## 5. Conclusions

This study provides novel insights into the population structure and genetic connectivity of *P. nausithous* in northern Spain. Genetic analyses demonstrate clear isolation between populations from Soria and those from the Cantabrian Mountains, suggesting limited gene flow across large geographical distances. Within the Montaña Palentina Natural Park, we identified a well-defined metapopulation in the Carrión River watershed that extends into the eastern Leonese mountains, suggesting that suitable habitat patches in this area effectively support butterfly movement and genetic exchange. However, the connectivity patterns in the Pisuerga River watershed remain uncertain due to sampling limitations, highlighting the need for additional research in this region. While our genetic analyses revealed no immediate extinction risk through population bottlenecks, the observed low genetic variability in Palencia populations warrants conservation attention. To enhance population resilience, we recommend maintaining and restoring existing habitat patches, protecting potential stepping-stone habitats between populations, and implementing management strategies that promote genetic exchange with neighboring populations. These measures would help increase heterozygosity within populations and reduce inbreeding risks, ultimately improving the conservation status of this vulnerable species. Future research should expand sampling efforts across the species’ range and employ multi-scale analyses to better understand connectivity patterns with neighboring populations, particularly in the eastern watershed region.

## Figures and Tables

**Figure 1 insects-16-00193-f001:**
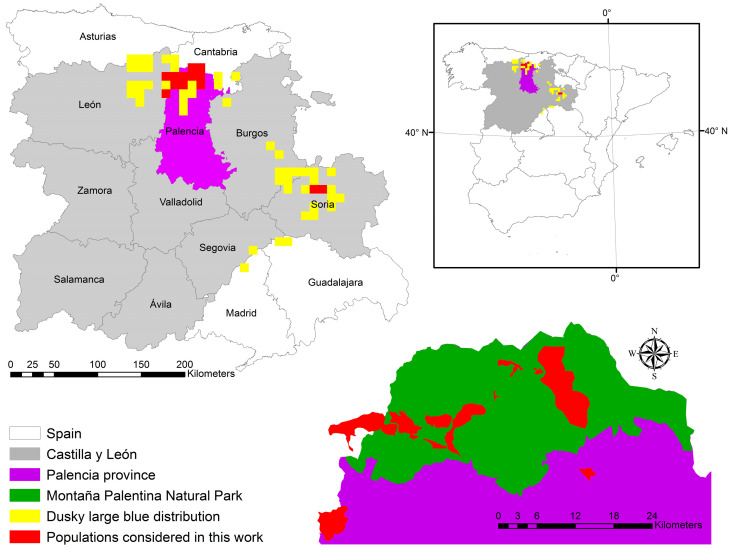
Location of the main study area (Palencia province) and distribution of *Phengaris nausithous* in Spain by 10 × 10 km UTM grid squares. The inset in the lower right corner provides a more detailed view of the Montaña Palentina Natural Park area and the study populations.

**Figure 2 insects-16-00193-f002:**
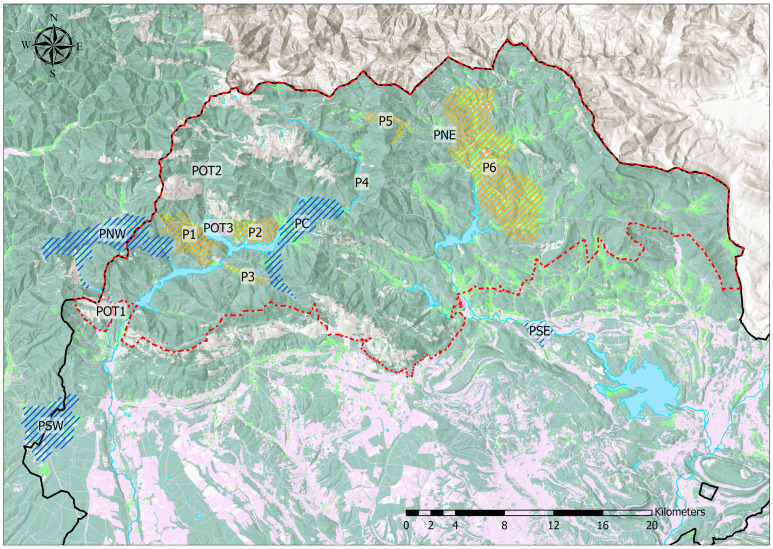
Distribution of *P. nausithous* populations in the Montaña Palentina Natural Park (boundaries in red) and surrounding areas overlaid on land cover and relief maps. Blue: genetically sampled populations described in Table 1 (PSW, PNW, PC, PNE, PSE), with PC, PNE, and part of PNW located within the Natural Park; orange: known but non-sampled populations detailed in Table 2 (P1–P6, all within the Natural Park); purple: patches with presence of the host plant *Sanguisorba officinalis* but no butterfly occurrence (POT1–POT3). The western part (Carrión River watershed) shows more rugged terrain, while the eastern part (Pisuerga River watershed) is characterized by gentler slopes and wider valleys. In the land cover map, dark green represents shrubland and forest areas; light green represents grasslands; grey represents bare rock; purple represents croplands; and blue indicates rivers, reservoirs, and lakes.

**Figure 3 insects-16-00193-f003:**
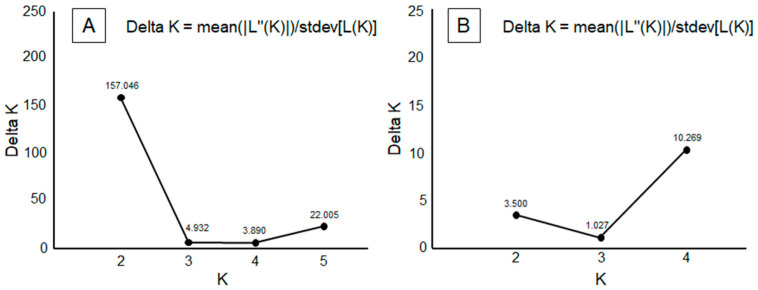
Delta *K* plot showing the average likelihood for each *K* based on 10 runs. (**A**) Δ*K* values as a function of *K*, showing the number of putative *P. nausithous* populations among six analyzed populations. (**B**) Δ*K* values as a function of *K*, showing the number of putative *P. nausithous* populations among five analyzed populations, excluding Soria.

**Figure 4 insects-16-00193-f004:**
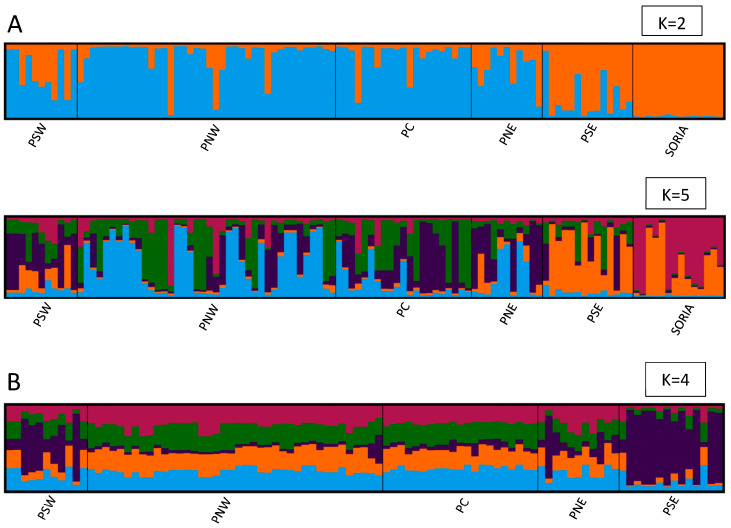
Population structure of butterflies determined by STRUCTURE: averages of 10 structure runs for each *K* value where each individual is represented by a vertical bar divided into *K* colors. *K* represents the assumed number of clusters. Each color represents a cluster, and the length of the colored segment indicates the estimated ancestry proportion of the individual for that cluster. Vertical black bars separate populations. (**A**) Population structure *K* = 2 and *K* = 5 for 111 butterflies from 6 populations (Soria, PSW, PNW, PC, PNE, and PSE). (**B**) Population structure *K* = 4 for 97 butterflies from 5 populations (excluding Soria).

**Table 1 insects-16-00193-t001:** Characteristics of the zones sampled for genetic analysis of *Phengaris nausithous* populations.

Population	Total Number of Patches	Sampled Patches	Area of Sampled Patches (ha)	Area of All Patches (ha)	Area of Matrix + Patches (ha)	Maximum Distance Between Patches (km)
PNW	6	3	2.96, 8.03, 10.2	23.78	2295	10
PC	8	2	3.78, 0.56	27.24	1487	8
PNE	1	1	1.96	1.96		
PSW	3	1	12.95	70.35	1668	3.8
PSE	1	1	3.20	3.20		
Soria	2	2	1.5, 6.8	8.3		2.6

**Table 2 insects-16-00193-t002:** Characteristics of non-sampled zones with *P. nausithous* presence (P1–P6).

Population	Total Number of Patches	Area of All Patches (ha)	Area of Matrix + Patches (ha)	Maximum Distance Between Patches (km)
P1	2	8.96	1059	2
P2	4	14.99	670	2.8
P3	1	18.88		
P4	1	1		
P5	1	6		
P6	25	186	5077	13

**Table 3 insects-16-00193-t003:** Population genetic parameters for *P. nausithous*. N: number of specimens; He: expected heterozygosity; Ho: observed heterozygosity; MNA: mean number of alleles; *Ar*: allelic richness; *F*is: inbreeding coefficient.

Locations	N	He	Ho	MNA	*Ar*	*F*is (1000 Bootstraps)
Soria	14	0.53	0.44	5.40	5.08	0.202
PSW	11	0.25	0.24	2.60	2.60	0.085
PNW	40	0.43	0.32	3.80	3.26	0.281
PC	21	0.29	0.24	2.40	2.24	0.204
PNE	11	0.37	0.27	2.80	2.80	0.312
PSE	14	0.40	0.46	2.80	2.75	−0.117

**Table 4 insects-16-00193-t004:** Genetic distances (upper diagonal) and gene flow (*N*m) values (lower diagonal) between populations.

Locations	Soria	PSW	PNW	PC	PNE	PSE
Soria	-	0.27	0.24	0.31	0.20	0.10
PSW	0.69	-	0.14	0.05	0.09	0.10
PNW	0.82	1.58	-	0.07	0.10	0.14
PC	0.56	3.51	4.60	-	0.10	0.15
PNE	1.00	2.30	2.44	2.27	-	0.07
PSE	2.29	1.59	2.28	1.37	3.18	-

**Table 5 insects-16-00193-t005:** Inter-patch distances (km) between sampled populations. The closest distances are marked with an asterisk (*), and the furthest distances with a dagger (^†^). Distance to Soria (not shown) was approximately 200 km for all Montaña Palentina patches (ranging from 178 to 214 km).

	PNW 1	PNW 2	PNW 3	PC 1	PC 2	PSW	PNE	PSE
PNW 1	-	2.5 *	10	9.2	14.3	18.8	25.1	30.7
PNW 2	2.5 *	-	8.1	11.5	15.9	19.2	26.4	33
PNW 3	10	8.1	-	19.2	23.9	16.2	34.3	40.8
PC 1	9.2	11.5	19.2	-	7.7	24	18.5	21.6
PC 2	14.3	15.9	23.9	7.7	-	31.4	11	19.5
PSW	18.8	19.2	16.2	24	31.4	-	42.3 ^†^	42
PNE	25.1	26.4	34.3	18.5	11	42.3 ^†^	-	17.8
PSE	30.7	33	40.8	21.6	19.5	42	17.8	-

## Data Availability

All of the data supporting the findings of this study are available within the paper. Should any raw data files be needed in another format, they are available from the corresponding author upon reasonable request.

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
