# Peer review of "Genetic Diversity and Connectivity of the Vulnerable Species *Phengaris nausithous* in Palencia (Northern Spain) [Author-notes fn1-insects-16-00193]"

_insects, 2025, doi:10.3390/insects16020193_

Round 1
Reviewer 1 Report
Comments and Suggestions for Authors
The submitted MS characterize genetic structure of endangered butterfly species, Phengaris nausithous from northern Spain, using microsatellite analysis. Authors presented interesting and original data, undoubtedly worthy for publication in “Insects” special issue. I do not see any moments that require improvements or corrections. Personally, I like the “Discussion” section, which is very comprehensive and based on a great number of literature sources and proper citations. I believe the MS can be accepted for publication.
Minor comment:
Please, italicize Wolbachia (line 374).
Author Response
Comments and Suggestions for Authors
The submitted MS characterize genetic structure of endangered butterfly species, Phengaris nausithous from northern Spain, using microsatellite analysis. Authors presented interesting and original data, undoubtedly worthy for publication in “Insects” special issue. I do not see any moments that require improvements or corrections. Personally, I like the “Discussion” section, which is very comprehensive and based on a great number of literature sources and proper citations. I believe the MS can be accepted for publication.
Q1. Minor comment: Please, italicize Wolbachia (line 374).
Response: We sincerely appreciate the reviewer’s positive assessment of our manuscript and their recognition of the comprehensive discussion supported by appropriate literature citations. We have addressed the minor comment by italicizing Wolbachia in line 374 as requested.
Reviewer 2 Report
Comments and Suggestions for Authors
The manuscript is well-written, and it makes a very interesting reading. The study has been properly executed and raises no major methodological concerns. While the results of its genetic analyses themselves do not seem particularly novel, they provide highly useful conclusions for practical conservation purposes as well as insightful new information about the persistence of a flagship butterfly species at its southern distribution limit. Nevertheless, I have a number of minor suggestions for the improvement of the study presentation – see below.
l. 71-75: The paragraph tends to mix ‘living dead’ populations with stepping stone habitats, which brings unnecessary confusion. In particular, while the term of ‘living dead’ is definitely relevant for the subject of the study, it is likely to be unfamiliar for a majority of readers, and thus nothing more than a catchy phrase for them. Hence, introducing it would only make sense with a proper explanation, for which there is not enough space in the Introduction. Nevertheless, it is definitely worth to be introduced and thoroughly considered in the Discussion, especially that Phengaris butterflies often persist in living dead populations as demonstrated by some earlier studies.
l. 80-81: The fragment “with only the more generalist species often remaining [10]” is hard to understand, but anyway it appears surplus, so I suggest deleting it.
l. 89: No reference is provided for the reported wingspan of the focal species, so the source of this information remains unclear. Is it the European butterfly database of Middleton-Welling et al. (2020, Scienfitic Data)?
l. 90 (and 2-3 other cases): You mostly write “host plant” in the text, so be consistent with the terminology and avoid using “food plant”.
Figs 1-4: In general, I have the impression that there are too many figures in the manuscript, possibly because the authors are just very fond of maps. In particular, I strongly believe that Figs 1 and 2 could better be combined into a single one, with the latter replacing the Palencia province map currently included in Fig. 1, whereas Fig. 3 is redundant as Fig. 4 shows the relief as well.
l. 123-130: This paragraph does not all fit it in here, and it should rather be moved to the Study Area section after line 164.
l. 159: “P. nausithous, which occurs in the first half of July” – later (l. 239-240 and l. 313-314) you indicate a much longer adult occurrence season, spanning July and August, so please correct the information given here.
l. 315-316: This information should rather belong to the Study Area Survey section than to the Results.
l. 392: “continuing with the analysis of population genetic structure” – remove this passage, which is absolutely unnecessary.
l. 397-398: “To explain these results, the literature describes aspects of both species biology and habitat matrix typology that may be key factors.” – this sentence need rephrasing as currently it reads very vague.
Table 5. There is no need to repeatedly show the same distance (> 200 km) to the external control population in Soria; it would be simpler to mention such information e.g. in this table heading.
l. 442: Change “even lower” to “also lower”, because it is not at all surprising that the genetic variability in Germany (the species core distribution areas) is higher than in the Cantabrian population, located at the species distribution margin.
l. 457-459: It is unclear why long-distance dispersal should be performed in particular by individuals with longer lifespans. This claim seems unsupported and highly speculative.
Reference #59: Please make sure that this reference is correct, because its title suggests that it is not on Phengaris nausithous.
Reference #59: “Banks, S.C.” is not an author for this reference, and must be removed.
